# Composites of N-butyl-N-methyl-1-phenylpyrrolo[1,2-a]pyrazine-3-carboxamide with Polymers: Effect of Crystallinity on Solubility and Stability

**DOI:** 10.3390/ijms241512215

**Published:** 2023-07-30

**Authors:** Vladimir B. Markeev, Evgenia V. Blynskaya, Sergey V. Tishkov, Konstantin V. Alekseev, Anna I. Marakhova, Alexandre A. Vetcher, Alexander Y. Shishonin

**Affiliations:** 1V.V. Zakusov Research Institute of Pharmacology, 8 Baltiyskaya St., 125315 Moscow, Russia; evaureus@gmail.com (E.V.B.); sergey-tishkov@ya.ru (S.V.T.); convieck@yandex.ru (K.V.A.); 2Institute of Biochemical Technology and Nanotechnology, Peoples’ Friendship University of Russia n.a. P. Lumumba (RUDN), 6 Miklukho-Maklaya St., 117198 Moscow, Russia; agentcat85@mail.ru; 3Complementary and Integrative Health Clinic of Dr. Shishonin, 5, Yasnogorskaya St., 117588 Moscow, Russia; ashishonin@yahoo.com

**Keywords:** amorphous solid dispersions, composite, solubility, bioavailability, anxiolytics, antidepressants

## Abstract

This work aimed to develop and characterize a water-soluble, high-release active pharmaceutical ingredient (API) composite based on the practically water-insoluble API N-butyl-N-methyl-1-phenylpyrrolo[1,2-a]pyrazine-3-carboxamide (GML-3), a substance with antidepressant and anxiolytic action. This allows to ensure the bioavailability of the medicinal product of combined action. Composites obtained by the method of creating amorphous solid dispersions, where polyvinylpyrrolidone (PVP) or Soluplus^®^ was used as a polymer, were studied for crystallinity, stability and the release of API from the composite into purified water. The resulting differential scanning calorimetry (DSC), powder X-ray diffractometry (PXRD), and dissolution test data indicate that the resulting composites are amorphous at 1:15 API: polymer ratios for PVP and 1:5 for Soluplus^®^, which ensures the solubility of GML-3 in purified water and maintaining the supercritical state in solution.

## 1. Introduction

The original active pharmaceutical ingredient GML-3 (Figure 1) synthesized in the FSBSI Zakusov Research Institute of Pharmacology (Russia, Moscow) has a high affinity (Ki = 5.3 × 10^−7^ M) with the 18 kDa mitochondrial translocator protein (TSPO) [1,2]. GML-3 shows pronounced antidepressant and anxiolytic activity when tested on animals, however, hydrophobicity and crystallinity are problems in the development of a dosage form for oral use [3].

Approximately 70% of new molecules developed as APIs are poorly soluble in water, resulting in limited dissolution and may adversely affect bioavailability [4,5]. GML-3 is practically insoluble in water (less than 1:10,000) and belongs to BCS class IV. Methods for increasing solubility include particle size reduction, API and polymer composites, complexing with cyclodextrins, generation self emulsifying systems (SEDDS) or liquid crystals (LCs), etc. Binary composite composites are a eutectic mixture or amorphous solid dispersion [6,7]. These composites make it possible to achieve a high level and rate of API dissolution.

A eutectic mixture composite is created by co-melting or dissolving a hydrophobic crystalline API and a semi-crystalline polymer (typically polyethylene glycol 1500, 4000, or 6000), followed by cooling or removal of the solvent, resulting in an API-PEG composite with a shifted down melting point with respect to API [6]. Thermal analysis (DSC) and powder X-ray diffraction (PXRD) are commonly used to assess the change in API crystallinity [7]. API solubility is usually evaluated in aqueous solutions (at different pH). Composite production will be indicated by the overall melting point for the API-PEG system. A partial flattening of the melting peak on DSC is possible due to the formation of a small number of poorly ordered molecular structures inside the crystal, which can be detected on PXRD. However, due to the preserved, even if altered, crystallinity of the composite, the increase in API solubility when released into the aquatic environment can be time-limited, as noted in a large number of studies. The reason for this is the inability of the polymer to create and maintain a saturated aqueous environment that ensures the solubility of the API crystals. Thus, API, due to its hydrophobicity and free energy acquired as a result of creating a composite with PEG, in an aqueous medium begins to precipitate over time or not dissolve at all, since the polymer does not prevent the agglomeration of API particles. As a result, the process is often poorly controlled and inefficient. Moreover, polymorphic forms of API, which are undesirable in the development and production of medicinal products, are difficult to identify in the case of composites obtained by this method, and an increase in solubility can be mistakenly perceived as the effect of PEG rather than polymorphism [8,9]. For this reason, it is advisable to create composites with amorphous polymers, where API molecules are separated from each other at the molecular level, thereby forming an amorphous solid dispersion.

The generation of amorphous solid dispersions is a modern approach to the creation of a composite that solves the problem of solubilization and its maintenance in an aqueous medium since the solubility of crystalline API can be increased by converting it into an amorphous form, which has a disordered arrangement of molecules over long distances and higher free energy along compared with its crystalline counterpart [10]. However, the excess free energy of the amorphous API can promote its recrystallization, thereby nullifying the advantages of the amorphous form [11,12,13,14,15].

Due to the higher free energy, physical stability is a major concern in the research and development of amorphous composites (ACs). Although thermodynamically stable ACs are preferred, most of the resulting composites are metastable due to the extremely low solubility of crystalline API in the polymer at room temperature, which often forces the concentration of the polymer to be increased to 1:5–1:20 relative to the water-insoluble substance and analyze the dynamics of the appearance of residual crystallinity in ACs by the DSC method [16,17,18]. This is because, during the production and storage of ACs, API molecules tend to aggregate with the formation of API-rich zones, which leads to the restoration of crystallinity and disruption of the integrity of the composite structure [19,20].

For this reason, in order to stabilize ACs by thermodynamic and kinetic methods, it is important to select a suitable polymer [21,22,23,24,25,26,27]. From a thermodynamic point of view, the free energy of mixing is related to the entropy and enthalpy mixing coefficient based on the theory of the Flory-Huggins lattice. The process is determined by the enthalpy component since the entropy of mixing is always positive and helps the mixing of API and polymer [28,29,30]. The enthalpy of mixing can be negative or positive, i.e., promote or oppose the mixing process depending on the interactions between the medicinal product and the polymer. If there are strong interactions between the API and the polymer, the enthalpy of mixing tends to be negative. As a result, the mixing of API and polymer occurs more easily, and more physically stable ACs can be obtained [31,32,33,34,35].

Systems with stronger API-polymer interactions have higher blendability resulting in better physical stability. Polymers can also stabilize ACs using a kinetic approach. It is assumed that molecular mobility, including global mobility (α-relaxation) and local mobility (β-relaxation), have a significant effect on the crystallization of amorphous API, and polymers can increase the relaxation time of API molecules, thereby limiting molecular mobility and inhibiting crystallization [36,37].

The paper considers composites representing GML-3 amorphous solid dispersions (hereinafter referred to as GML-3 amorphous composite or AC-GML-3 for short) with water-soluble polymers, obtained by joint dissolution of GML-3 and water-soluble polymers in ethyl alcohol, followed by solvent removal. This technique was chosen because of the ease of use and scaling when transferring technology to production. It is also observed that prolonged thermal exposure causes a gradual visually observed darkening (from light yellow to black) in the GML-3 sample. Degradation is followed by the appearance of impurities identified by HPLC, in the amount exceeding the maximum permissible values for API. This observation prohibits to use of hot melt extrusion technology, and the risk of formation of other polymorphic forms with a decrease in crystallization time makes it undesirable to use freeze-drying or supercritical fluid technology. After recrystallization with ethanol, any changes in the crystal structure of GML-3 have not been revealed. It is also worth noting that GML-3 degrades under prolonged thermal exposure, which excludes the use of hot melt extrusion technology, and the risk of formation of other polymorphic forms with a decrease in crystallization time makes it undesirable to use freeze-drying or supercritical fluid technology. After ethanol recrystallization, no changes in the crystal structure of GML-3 were observed.

PVP (Kollidon^®^ 25) and Soluplus^®^ are highly soluble in both water and alcohol, making them the most suitable polymers for making AC-GML-3. Kollidon^®^ 25 is one of the most commonly used polymers for creating composites with optimal viscosity. According to some studies, Soluplus ^®^ can create weak bonds with API, improving its solubility. Kollicoat^®^ IR was chosen as the third polymer, however, due to its insolubility in alcohol, the dissolution was carried out in 50% ethanol. This made it possible to evaluate the possibility of using a mixture of solvents in the creation of AC-GML-3. The current study is the first systematic research devoted to the issue of poor solubility of GML-3 and the creation of composites as a way to solve this problem. The general aim was to improve the solubility of GML-3, by creating composites with polymers, which will allow the API to be transferred to a stable amorphous state and increase its solubility for the subsequent creation of an oral form of a drug with improved biopharmaceutical characteristics (at least without losses of its antidepressant and anxiolytic effect).

## 2. Results and Discussion

The samples were GML-3, mechanical mixtures of GML-3 with Soluplus^®^, PVP, or Kollicoat^®^ IR in the ratio of GML-3:polymer = 1:10, AC-GML-3 at the ratios of GML-3:polymer = 1:5, 10, 15, 20. To evaluate the change in GML-3 crystallinity in the composites, the obtained samples were analyzed by PXRD and DSC methods. It also made it possible to confirm the formation of the composite and its type. In the case of a eutectic mixture (possibly for Soluplus^®^ and Kollicoat^®^ IR, since these polymers contain PEG), the thermogram of the samples should differ from the data for GML-3 and the polymer separately, and PXRD should show peaks specific of crystalline (fully or partially) substances. In the case of amorphous solid dispersion, the thermogram should be characteristic of the vitreous substance. In the case of residual crystallinity, smoothed endothermic peaks will be present on the thermogram. In addition to the glass transition of the composite, the melting of residual GML-3 crystals will occur simultaneously.The creation of a eutectic mixture with a semicrystalline polymer is not included in the goals of the study, since a relatively low solubility of GML-3 in water from the obtained composites and a rapid drop in concentration as early as 30 min after the start of the experiment were revealed.

According to PXRD and DSC data, GML-3 has a complex crystal structure with a melting point of 87 °C, which is retained upon repeated heating or cooling (Figure 2). Even though the GML-3 is cooled down in the device, the thermogram demonstrates no significant changes in comparison with heating, when checking the cooled sample by HPLC, 0.9% of impurities (at least 4 in different amounts) were detected. This is 4.5 times higher than the maximum permissible value for this API and 18 times the data for the original GML-3 before heating and explains the absence of changes on the thermogram and confirms the undesirability of using the extrusion method to create composites. Soluplus^®^ and PVP polymers are amorphous, with two specific halos.

### 2.1. Crystallinity Analysis of GML-3 Composites with Soluplus^®^

According to the PXRD data, the resulting GML-3-Soluplus^®^ composite was completely amorphous at all concentrations (Figure 3a). However, there is a decrease in the first amorphous halo (9°–13°) characteristic of the polymer, which is especially pronounced at a concentration of 1 to 15 (Figure 3a). Thus, according to the literature, there is an interaction between API and Soluplus^®^ [8,34]. For the same reason, there is a slight shift from 20° to 22° at a concentration of 1 to 15 and the appearance of a new amorphous halo at 40–45°. A similar situation is observed at a ratio of 1 to 20.

The DSC data demonstrates the presence of residual crystallinity in some of the samples with its gradual disappearance with an increase in the mass fraction of the polymer in the composite (Figure 3b). According to the thermogram data, the glass transition process of the composite and the melting process of residual crystals are observed. For this reason, for AC-GML-3 with 1:5 and 10, there is a wide peak in the 35–80 °C region on the thermogram.

For a ratio of 1 to 5, a pronounced melting effect is observed, despite the initial glass transition process. This is explained not by the presence of GML-3 crystallization zones, as a result of which AC-GML-3 glass transition and melting of residual GML-3 crystals occur simultaneously during heating, but by the properties of Soluplus^®^ itself (Figure 3b). The Soluplus^®^ polymer contains PEG, which causes a melting effect at 64 °C, and DSC shows a performance similar to 1 to 5 (if the peak shift from 64 °C to 62 °C for AC-GML-3 is taken into account). For ratios of 1 to 5, as the concentration increases to 1 to 10, residual crystallinity during glass transition is also observed, but the data are less typical for PEG, despite an increase in its mass fraction in AC-GML-3, which indicates interaction within the composite. For values from 1 to 15 and 20, the thermogram is characteristic of a glassy material. In other words, AC-GML-3 is completely amorphous and stable because intermolecular interaction is difficult due to the distance and the connection of the API with the polymer (Figure 3b).

The obtained samples of AC-GML-3 (Soluplus^®^ polymer) were re-examined after accelerated aging (30° above room temperature in a closed vessel for 139 h), resulting in no significant changes in DSC and PXRD, indicating GML-3 crystallization inside the composite. So, for example, AC-GML-3 at a ratio of 1 to 20 on DSC, after repeated examination, recorded changes in the amorphous state, however, not leading to the formation of crystals (Figure 3c,d).

### 2.2. Crystallinity Analysis of GML-3 Composites with Kollidon^®^ 25

According to the PXRD data, at a ratio of 1 to 5, a pronounced retention of the crystalline phase is observed in comparison with GML-3 (Figure 4f). However, partial amorphization occurs, since, in comparison with the mechanical mixture of PVP-GML-3, the intensity of the peaks is reduced, i.e., during mechanical mixing, all the main peaks of GML-3 on PXRD are preserved and well identified, and their intensity decreases due to crystal refinement and an amorphous halo created by the polymer (Figure 4e,f). When creating AC-GML-3 in a ratio of 1 to 5, crystallinity is also observed, but less pronounced, since GML-3 and Kollidon^®^ 25 were jointly dissolved in alcohol, as a result of drying this solution, GML-3 recrystallized, but only partially (Figure 4a,b). In other words, there are zones with crystalline GML-3 inside the sample.

At a ratio of 1 to 15 and 20, a pronounced increase in the intensity of the second halo characteristic of the polymer is observed (for PVP, the second and first halo are of the same intensity), which may indicate a weak interaction between GML-3 and PVP inside the composite (Figure 4a,c). Repeating the experiment many times retained the picture with the change in the second halo; moreover, an increase in intensity was also observed for 1 to 10, as shown in Figure 4c,d. To define possible variations, the experiment was carried out in 3 parallel samples. No differences, that could be caused by e.g., the lack of homogenety, were observed. However, it was in the case of 1 to 10 that, upon re-measurement after 3 months, the second halo dropped to the values characteristic of PVP, which may indicate the instability of the composite in this ratio (Figure 4c,d).

Given this circumstance, in addition to the usual study of samples for DSC, a study was carried out after accelerated aging (at a temperature above room temperature by 30 °C in a closed vessel for 46 and 139 h). The results are shown in Figure 5. Immediately after drying and obtaining a solid powder from a polymer solution and GML-3, all samples at all studied concentrations did not show the melting temperature characteristic of GML-3 crystals (Figure 5).

At a ratio of 1 to 5, traces of residual crystallinity are observed, remaining almost imperceptibly at 1 to 10, and disappearing further, at 1 to 15 and 20, the samples are completely amorphous. Sharp fluctuations after 110 °C are associated with sample effervescence and are not characteristic of powders. Thus, after drying the samples, no crystal structures characteristic of GML-3 were observed, but only signs of residual crystallinity.

After the test by the accelerated aging method, AC-GML-3 (1 to 5) quickly acquired crystallinity, and 1 to 10 decomposition appeared in the presence of the polymer and GML-3, at 1 to 15 they retained amorphousness and showed sufficient stability (Figure 5). Thus, at ratios of 1 to 5, 10, centers of crystallization of GML-3 or zones of increased API content are formed rather quickly, which further leads to a gradual restoration of the crystal structure of GML-3. With an increase in the content of PVP in AC-GML-3 (up to 1 to 15 or more), the onset of processes preceding the crystallization of API was not detected during 139 h of accelerated aging. Thus, it can only be mentioned about the stability of the obtained composites with PVP only at a ratio of 1 to 15 or more.

The use of a water-alcohol mixture to create AC-GML-3 based on ethanol-insoluble Kollicoat^®^ IR turned out to be difficult since PXRD and visual control data showed complete retention of crystallinity and separation of GML-3 and polymer during drying. So, in contrast to Kollidon^®^ 25, Kolicoat^®^ IR is characterized by an amorphous halo (with a maximum intensity at an angle of 20), which does not overlap the GML-3 peak with the highest intensity, which made it possible to identify, without DSC, the presence in the sample of peaks characteristic of GML-3 crystals (Figure 6).

It can be concluded that after a comprehensive study of GML-3 composites with water-soluble polymers, there is a strong interaction between the GML-3 and Soluplus^®^ molecules, which synergistically improves the stability and amorphism of API, with a completely amorphous state ranging from 1 to 5. For PVP, instability of ACs was revealed at ratios up to 1 to 15 (not included) with the return of crystallinity, and weak polymer-GML-3 interaction compared to Soluplus^®^. The use of Kolicoat^®^ IR to create a composite proved to be impractical. To assess the degree of release of GML-3 from composites into purified water, a comparative dissolution kinetics of AC-GML-3 in comparison with GML-3 was carried out using the “dissolution” test.

### 2.3. Comparative Dissolution Kinetics of GML-3-PVP and GML-3-Soluplus^®^ Composites

According to the “Dissolution test for solid dosage forms” carried out with composites, there is an increase in the level and release rate for composites with PVP or Soluplus^®^ compared to GML-3 and mechanical polymer blend with GML-3, ranging from a concentration of 1 to 5 (Figure 7). At concentrations above 1 to 15, a slowdown in the dissolution rate is observed due to the large amount of polymer used. It should be noted that when repeating the experiment with the same samples after 12 months for PVP, there was a slight decrease in the release level for 1 to 5 and 10 (by 25–30%), which indicates the instability of ACs at these concentrations, as previously demonstrated by the DSC method (with accelerated aging).

It should be noted that the high level of release of GML-3 from composites indicates its promise as the basis for creating an oral dosage form (tablets), however, the technology for obtaining GML-3 tablets based on the GML-3-PVP or GML-3-Soluplus^®^ composite requires additional research.

As a result, by the solvent removal method, it was possible to obtain stable GML-3 composites with Soluplus ^®^ or PVP and also to exclude Kolicoat ^®^ IR as a polymer that prevents the crystallization of GML-3. Soluplus ^®^ forms a weak molecular bond with GML-3 in the composite (PXRD data), for PVP, signs of weak interaction were observed only at ratios of 1:5.10. The PXRD and DSC data made it possible to prove the transition of GML-3 from a crystalline state to an amorphous one (in a composite) and to select the most stable composites (after 138.5 h at a temperature of 55 °C). The data on the dissolution kinetics given at the last stage allow us to conclude that it is optimal to use composites with GML-3 to Soluplus ^®^ at a ratio of 1:5, GML-3 to PVP at a ratio of 1:15. Namely at these ratios of GML-3 and polymer the highest solubility/dissolution rate of API was observed.

## 3. Materials and Methods

### 3.1. Materials

GML-3 was obtained from FSBSI Zakusov research institute of pharmacology (Moscow, Russia). Release series: 30082021, manufactured on 30 August 2021. Figure 1 shows the structural formula of GML-3, and Table 1—its physical and chemical parameters.

Excipients (Es): Polymer polyvinylpyrrolidone (Kollidon^®^ 25), graft copolymer polyvinyl caprolactam-polyvinyl acetate polyethylene glycol (Soluplus^®^), and polyvinyl alcohol-polyethylene glycol graft copolymer (Kollicoat^®^ IR) were obtained from (BASF AG, Ludwigshafen, Germany).

Solvents: distilled water was obtained on the PE-2205 apparatus (Ecroskhim Ltd., St. Petersburg, Russia), and ethanol HP (99.5%, 0.005% maximum water) was obtained from (Merck KGaA AG, Darmstadt, Germany).

### 3.2. Methods

#### 3.2.1. Generation of AC-GML-3

GML-3 (1.00 g) was loaded into a closed container (100 mL) and dissolved in 99% ethanol (35.00 g). The dissolution was carried out using a magnetic stirrer PE-6100 (Ecroskhim Ltd., St. Petersburg, Russia). Kollidon^®^ 25 or Soluplus^®^ was gradually added to the resulting solution and mixed until a homogeneous transparent solution was obtained. The resulting solution was dried at 55 °C for 12 h. In the case of using Kollicoat^®^ IR as a polymer, 35 g of an aqueous solution of Kollicoat^®^ IR was gradually added with stirring to the alcoholic solution of GML-3, followed by drying for 24 h at a temperature of 55 °C.

#### 3.2.2. Generation of a Mechanical Mixture of GML-3 and Polymer

For comparison, in the “Dissolution test for solid dosage forms” and PXRD tests, a mechanical mixture of GML-3 and polymer was used, obtained by joint grinding of API (20 mg) and polymer (200 mg) in an agate mortar for 20 min.

#### 3.2.3. DSC

Measurement of AC-GML-3 by DSC was performed using a Netzsch STA 449 F1 (Netzsch Instruments North America LLC, Burlington, MA, USA) combined with a QMS 403 C thermogravimetry. Approximately 5 mg of the sample was weighed and sealed in an aluminum container with a hole in the lid. The samples were heated to 160 °C at a heating rate of 5 °C/min. Air was used as the purge gas.

This method allows fixing the heat of fusion/glass transition of composites. The endothermic peak is characteristic of crystalline substances. Smoothed drops without a pronounced peak are characteristic of the glass transition process, which indicates the amorphous nature of the substance. Thus, this technique is suitable for assessing the crystallinity of the obtained samples.

#### 3.2.4. PXRD

The PXRD spectra of AC-GML-3 were recorded on a desktop X-ray diffractometer a Miniflex 600 (Rigaku Corp., Tokyo, Japan). The measurements were carried out Cu Ka radiation at 40 kV and 40 mA in the range 2θ 2–60° with a scanning speed of 4°/min and a step size of 0.02°.

The method allows to estimate the degree of crystallinity of the substance. Each crystal lattice is characterized by a certain set of peaks. The appearance of a halo on the diffractogram indicates the presence of an amorphous substance. Diffractograms of the studied composites can be halos with/without peaks characteristic of GML-3. The polymer used to create the composite will give an amorphous halo, and the API, if the crystallinity is preserved, will give peaks characteristic of the crystal. The absence of peaks of crystalline GML-3 on diffractograms will indicate the production of amorphous composites.

#### 3.2.5. Dissolution Test

Carried out according to European Pharmacopoeia 10 edition (2.9.3. Dissolution test for solid dosage forms). The medium of dissolution was distilled water with a volume of 900 mL. Mixing was carried out using a paddle mixer at a rate of 50 rpm. Dissolution medium temperature: 37.0 ± 0.5 °C; Sampling time: at 1, 3, 5, 10, 15, 30, 45, 60 min; Replenishing the medium after each sampling (10 mL). The optical density was determined using a spectrophotometer PE-5400UF (Ecroskhim Ltd., St. Petersburg, Russia) at a wavelength of λ = 256 nm.

This method is appropriate for the evaluation of the kinetics of dissolution of GML-3 and AC-GML-3. An increase in the solubility of the API according to the BCS may indicate an improvement in the bioavailability of the API.

#### 3.2.6. Accelerated Aging Stability Tests

AC-GML-3 samples were stored in closed containers at 55 ± 0.5 °C for 46 and 139 h, which approximately corresponds to 1 and 3 months of storage at room temperature.

## 4. Conclusions

The results of the study show that the creation of a composite of GML-3 with PVP or Soluplus^®^ affects the solubility, both due to the solubilizing properties of water-soluble polymers (increasing the solubility of the mechanical mixture), and by changing the crystallinity of GML-3. At a minimal ratio of GML-3 to Soluplus ^®^ (1:5) and PVP (1:15), the composite retains its stability for a long time without re-crystallization. The resulting composites make it possible to achieve a high level of release of GML-3 into purified water and thereby ensure the bioavailability of a medicinal product (antidepressant, anxiolytic) of the combined action.

## Figures and Tables

**Figure 1 ijms-24-12215-f001:**
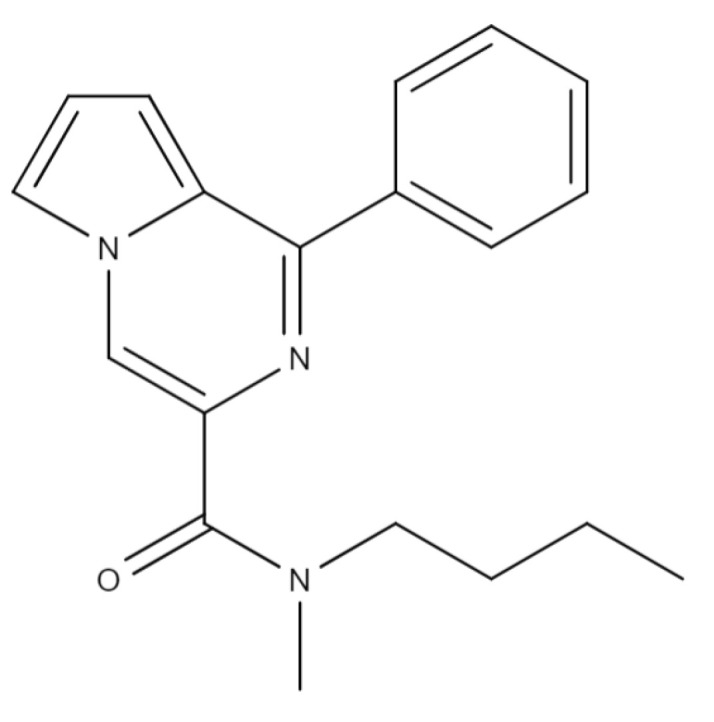
Structural formula of GML-3.

**Figure 2 ijms-24-12215-f002:**
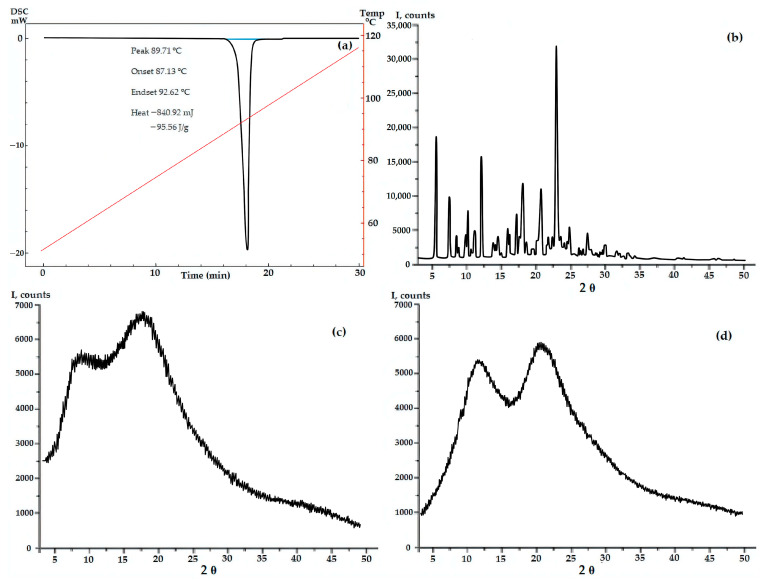
DSC data for GML-3 (**a**) and PXRD data for GML-3 (**b**), Soluplus^®^ (**c**) and Kollidon^®^ 25 (**d**).

**Figure 3 ijms-24-12215-f003:**
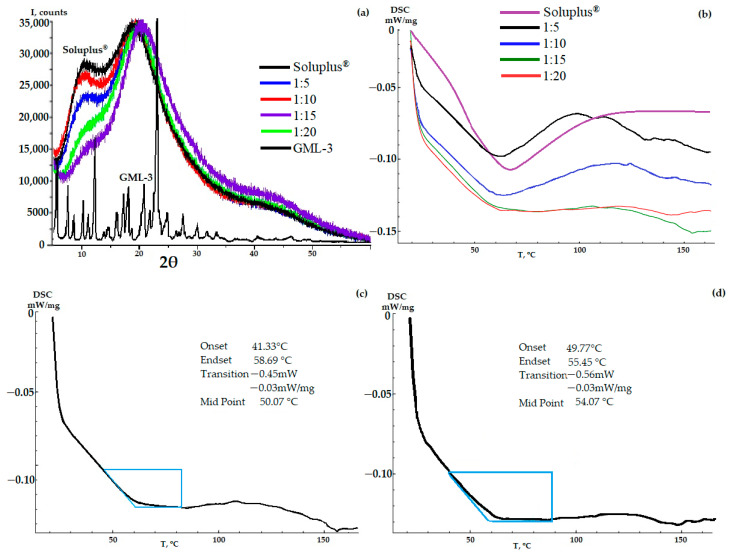
PXRD (**a**) and DSC (**b**) data of AC-GML-3 (based on Soluplus^®^) at various concentrations, DSC of AC-GML-3 samples (Soluplus^®^ polymer) 1 to 15 and 20 after accelerated aging (**c**,**d**) respectively.

**Figure 4 ijms-24-12215-f004:**
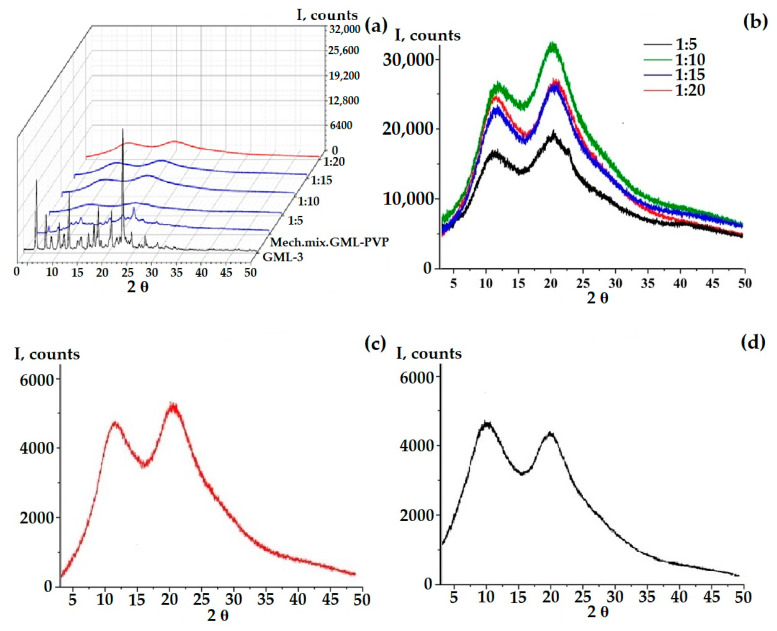
PXRD data of samples: AC-GML-3 (based on Kollidon^®^ 25) at various concentrations, GML-3 substance, and a mechanical mixture of GML-3 and PVP (1:10) (**a**,**c**), AC-GML-3 (polymer Kollidon^®^ 25) at a ratio of 1 to 10, immediately after drying of AC-GML-3 (**b**) and after 3 months of storage (**d**), a mechanical mixture of GML-3 and Kollidon^®^ 25 at a ratio of 1:5 (**e**,**f**).

**Figure 5 ijms-24-12215-f005:**
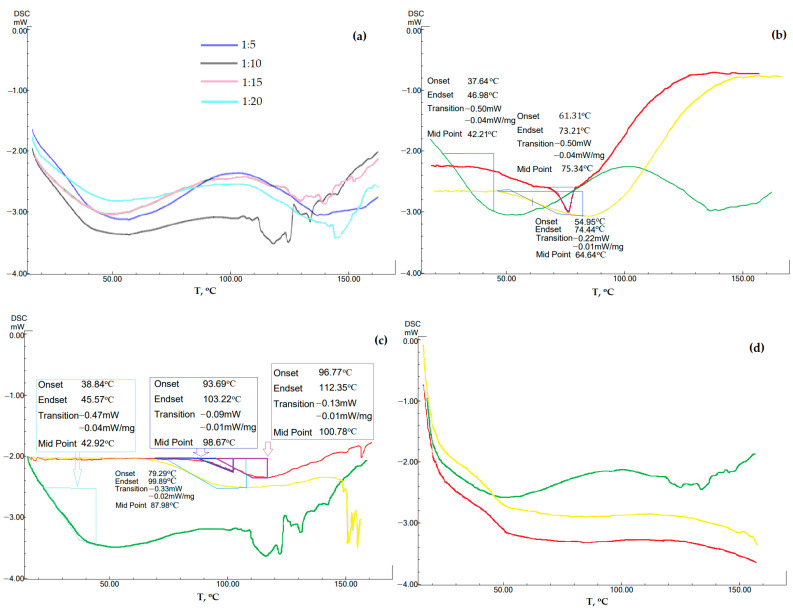
Change in the degree of crystallinity of AC-GML-3 (polymer Kollidon^®^ 25) by DSC of samples after accelerated aging: thermogram of AC-GML-3 at different concentrations of Kollidon^®^ 25 (**a**); API: PVP = 1:5 right after drying (green), 46 h after drying (yellow), and 139 h after drying (red) (**b**); API: PVP = 1:10 right after drying (green), 46 h after drying (yellow), and 139 h after drying (red) (**c**); API: PVP = 1:15 immediately right after drying (green), 46 h after drying (yellow), and 139 h after drying (red) (**d**).

**Figure 6 ijms-24-12215-f006:**
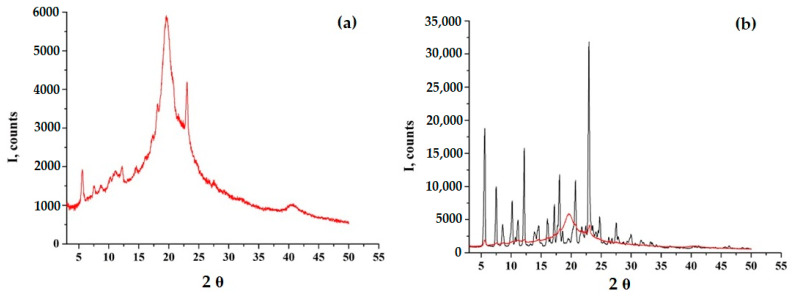
PXRD data of a mixture of GML-3 with Kolicoat^®^ IR after recrystallization in 50% ethanol at an API: Kolicoat^®^ IR ratio of 1:10 without GML-3 PXRD overlay (**a**) and with GML-3 PXRD overlay (**b**).

**Figure 7 ijms-24-12215-f007:**
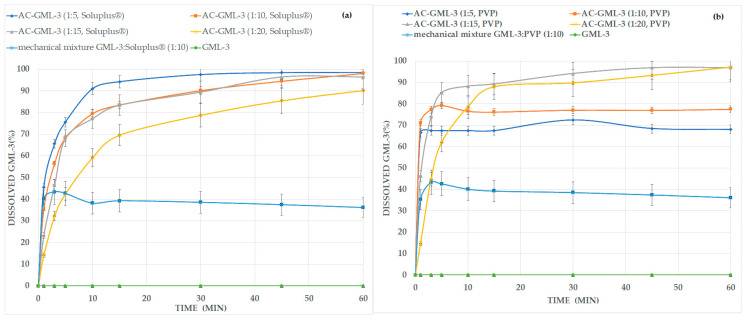
GML-3 release kinetics from API with Soluplus^®^ (**a**) and PVP (**b**) composite into purified water.

**Table 1 ijms-24-12215-t001:** Physicochemical properties of the GML-3 compound.

Properties	Value
Molecular mass (M, g/mol)	307.39
Melting temperature (Tm, °C)	87–89
Solubility in water	more than 1:10,000
Solubility in ethanol	1:7

## Data Availability

The data presented in this study are available on request from the corresponding author.

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
