# Peer review of "Composites of N-butyl-N-methyl-1-phenylpyrrolo[1,2-a]pyrazine-3-carboxamide with Polymers: Effect of Crystallinity on Solubility and Stability"

_ijms, 2023, doi:10.3390/ijms241512215_

Round 1
Reviewer 1 Report
I suggest the acceptance after some minor corrections as follows;
1. Rearrange the sentences in Abstract section
2. What is the role of polyvinylpyrrolidone in this study
3. Reformulate the aim of the work in introduction
4. What is the novelty of the work
5. Add previous published work with comparison to clear the novelty of your work
6. Add more explanation to experimental work
7. Figure 2a need to be replaced my another one more resolution
8. Figure 3 c, d need to be replaced my another one more resolution
9. Figure 7 a, b need to be replaced my another one more resolution
10. Give some results in conclusion part
11. Correct typographical errors.
12. What is the application of this study?
Author Response
To Reviewer 1
Dear Reviewer,
First of all, let me thank you for your evaluation of our masterpiece and efforts to make our submission the most attractive to MDPI readers. As about your comments, let me address them in the order of their appearance in your review:
- Rearrange the sentences in Abstract section
Done
- What is the role of polyvinylpyrrolidone in this study
Explained
- Reformulate the aim of the work in introduction
Done
- What is the novelty of the work
Explained
- Add previous published work with comparison to clear the novelty of your work
Done
- Add more explanation to experimental work
Done
- Figure 2a need to be replaced my another one more resolution
Done
- Figure 3 c, d need to be replaced my another one more resolution
Done
- Figure 7 a, b need to be replaced my another one more resolution
Done
- Give some results in conclusion part
Done
- Correct typographical errors.
Done. Moreover, we invited the native American speaker to polish the final version
- What is the application of this study?
Explained
Please let us know if we could do something else to improve the quality of our submission.
Regards
Alex
Dr. Alexandre A Vetcher
Head of the Nanotechnology lab at the
Scientific and educational center "Nanotechnologies"
at the Institute of Biochemical Technology and Nanotechnology (IBTN)
of Peoples' Friendship University of Russia (RUDN),
10/2 Miklukho-Maklaya Str, Moscow, 117198, Russian Federation
and
Associate Director (Science) at
Complementary and Integrative Health
Clinic of Dr. Shishonin
5 Yasnogorskaya Str, Moscow, 117588, Russian Federation
Reviewer 2 Report
1. The DSC was carried out in air. The authors said the GML will degrade if using the melting mixing method. So, did the GML degraded during the DSC testing?
2. The unit “ml” should be “mL”.
3. “Figures 5” should be “Figure 5” in line 270.
4. The Figures are not clear. Image resolution needs to be further improved.
Further improvement is needed in English writing. The manuscript is difficult to understand.
Author Response
To Reviewer 2
Dear Reviewer,
First of all, let me thank you for your efforts to add our submission the best shape. As about your comments, let me address them in the order of their appearance in your review:
- The DSC was carried out in air. The authors said the GML will degrade if using the melting mixing method. So, did the GML degraded during the DSC testing?
We explained it in the body
- The unit “ml” should be “mL”.
Done
- “Figures 5” should be “Figure 5” in line 270.
Done
- The Figures are not clear. Image resolution needs to be further improved.
We improved the quality and hope that now it looks much better
- Comments on the Quality of English Language: Further improvement is needed in English writing. The manuscript is difficult to understand.
We invited the native American speaker to polish the final version
Please let us know if we could do something else to improve the quality of our submission.
Regards
Alex
Dr. Alexandre A Vetcher
Head of the Nanotechnology lab at the
Scientific and educational center "Nanotechnologies"
at the Institute of Biochemical Technology and Nanotechnology (IBTN)
of Peoples' Friendship University of Russia (RUDN),
10/2 Miklukho-Maklaya Str, Moscow, 117198, Russian Federation
and
Associate Director (Science) at
Complementary and Integrative Health
Clinic of Dr. Shishonin
5 Yasnogorskaya Str, Moscow, 117588, Russian Federation
Reviewer 3 Report

Several sentences are long due to which the message is unclear. Recommend the authors avoid long sentences and focus on concise messaging.
Author Response
2023-07-19 to Rev.3
Dear Reviewer:
First of all, let me thank you for your efforts to make our submission better to readers. Let me respond to your comments in their order in your review:
- Several technical aspects of the article are not well articulated.
Improved accordingly
- There are several references to the results using the figures, which are not well explained.
Improved accordingly
3.The test methods used and their significance in the current context are not described. This is important as results of these tests are used to draw significant conclusions.
Improved
- Lines 38-41 unclear
Improved
- Lines 42-25 are unclear. Recommend concise rewriting
Improved
- Line 49: Unclear what “control of the resulting composite” means
Improved
- Lines 69-73, Lines 77-82, Lines 107-111, Lines 112-117, Lines191-193, Lines 264-267: Need concise sentences. Message unclear.
Improved accordingly
- Not enough discussion on existing methods to increase solubility of API
Improved
- Line 210: Where are the repeated heating and cooling occurring in Figure 2?
Improved and corrected
- Lines 210-216, 316-317: Unclear sentences.
Improved accordingly
- Sections 2.2.3, 2.2.4, 2.2.5: Recommend adding what is the purpose of these tests, how are these tests performed and what to look for in these test results. This is important as later in the discussion results of these tests are discussed and used to draw conclusions.
Improved accordingly
- Line 224: Slight shift compared to which plots. Recommend identifying the plots to assist user in locating the shift.
Improved accordingly
- Line 200: What do the authors mean by “without the parallel presence of the melting process characteristic”
Improved accordingly
- Line 233-235: How does the DSC plot provide information on the % of crystallinity?
Explanation added
- Lines 240-242: which plots are these shown in?
Improved accordingly
- Lines 246-250: Unclear sentence
Improved
- Lines 253: Did the authors mean absence of recrystallization inside the composite?
Yes. Improved accordingly
- Lines 259-260: Recommend identifying this in the plot
Improved accordingly
- Line 284-286: Why was the experiment repeated many times?
Improved accordingly
- Lines 364-367: Which aspect of the result supports the conclusion?
Improved accordingly
Please let us know if we can do something else to improve the quality of our submission.
Sincerely
Dr. Alex Vetcher
Reviewer 4 Report
Dear Authors,
It is interesting AC-GML-3 solubility improvement through amorphous solid phase.
However need to improve data.
Need to write what class of BCS and solubility studies in details.
DSC experiments are not presented in clear way and Figure 5 units of temperature is not correct in the figure.
PXRD images need to improve the quality.
Can you add drug solubility reported by different methods and tell how novel it is now ?
Next the stability studies of amorphous form not done ? how many months or years this amorphous phase is stable ? id this will go to crystalline phase what phase it is ?
Did you do VTXRPD ?
Need to make clear images with good quality.
I suggest for major revision
Author Response
2023-07-19 to Rev.4
Dear Reviewer:
First of all, let me thank you for your efforts to make our submission better to readers, as well as for high evaluation of the obtained data. Let me respond to your comments in their order in your review:
- Need to write what class of BCS and solubility studies in details.
Added
- DSC experiments are not presented in clear way and Figure 5 units of temperature is not correct in the figure.
Improved
- PXRD images need to improve the quality.
Improved
- Can you add drug solubility reported by different methods and tell how novel it is now ?
Partially added
- Next the stability studies of amorphous form not done ? how many months or years this amorphous phase is stable ? id this will go to crystalline phase what phase it is ?
According to 2 years of storage (at room temperature) compositions with PVP and Solo plus in ratios of 1:15 and 1:20 retained amorphousness. In compositions 1:5 and 1:10, in addition to the restoration of crystallinity after two years, a large number of impurities were identified (presumably from the decomposition of GML-3). Research in this area continues and will be reported after obtaining the final data.
- Did you do VTXRPD ?
No. Based on the DSC, it was assumed that no other polymorphic forms of GML-3 are formed when heated. The issue of obtaining polymorphic forms of GML-3 was considered in another study that is being prepared for publication. In the study, it was possible to obtain a polymorphic form of GML-3 by one of the recrystallization methods. VTXRPD was not carried out for composites, since the temperature dependence of their crystallinity on temperature (in the range from 20 to 86℃) is unlikely.
- Need to make clear images with good quality.
Improved
- I suggest for major revision
We tried our best.
Please let us know if we can do something else to improve the quality of our submission.
Sincerely
Dr. Alex Vetcher
Round 2
Reviewer 2 Report
The authors have carefully revised the paper and it basically meets the requirements for publication.
Some sentence still can be improved, such as "This is 4.5 times higher than the maximum permissible value for this API and 18 times the data for the original GML-3 before heating and explains the absence of changes on the thermogram and confirms the undesirability of using the extrusion method to create composites." The sentence is too long and difficult to be understand.